# Twinning Behavior in Cold-Rolling Ultra-Thin Grain-Oriented Silicon Steel

**Bo Zhang** [1], **Li Meng** [1,*], **Guang Ma** [2], **Ning Zhang** [1], **Guobao Li** [3], **Kun Liu** [4] **and Sheng Zhong** [5]

[1] Metallurgical Technology Institute, Central Iron and Steel Research Institute, Beijing 100081, China; zhangbo530925@163.com (B.Z.); zhangning@cisri.com.cn (N.Z.)

[2] State Key Laboratory of Advanced Power Transmission Technology, Global Energy Interconnection Research Institute Co., Ltd., Beijing 102211, China; maguang@geiri.sgcc.com.cn

[3] Baoshan Iron and Steel Co., Ltd., Shanghai 201900, China; ligb@baosteel.com

[4] China Three Gorges Corporation, Beijing 100038, China; liu_kun@ctg.com.cn

[5] ABB Corporate Research Center, 940 Main Campus Dr, Suite 200, Raleigh, NC 27606, USA; szhongud@gmail.com

* Correspondence: mengl@cisri.com.cn or li_meng@126.com

**Abstract:** Twinning behaviors in grains during cold rolling have been systematically studied in preparing ultra-thin grain-oriented silicon steel (UTGO) using a commercial glassless grain-oriented silicon steel as raw material. It is found that the twinning system with the maximum Schmid factor and shear mechanical work would be activated. The area fraction of twins increased with the cold rolling reduction. The orientations of twins mainly appeared to be $\alpha$-fiber (<110>//RD), most of which were {001}<110> orientation. Analysis via combining deformation orientation simulation and twinning orientation calculation suggested that {001}<110> oriented twinning occurred at 40–50% rolling reduction. The simulation also confirmed more {100} <011> oriented twins would be produced in the cold rolling process and their orientation also showed less deviation from ideal {001}<110> orientation when a raw material with a higher content of exact Goss oriented grains was used.

**Keywords:** ultra-thin grain-oriented silicon steel (UTGO); cold rolling; twinning; Goss; {100}<011>

## 1. Introduction

Ultra-thin grain-oriented silicon steel (UTGO steel, thickness ≤ 0.10 mm) is an important magnetic material mainly used for manufacturing intermediate and high-frequency transformers [1–4], thanks to its ability to increase core power while reducing core loss and volume. Although the manufacturing route for grain-oriented silicon steel has been developed for decades, it is still difficult to produce ultra-thin products using a conventional process which is based on secondary recrystallization. The difficulty is due to acceleration of the inhibitor coarsening during recrystallization and poor control over Goss orientation under large rolling reduction [5–9]. At present, the most prevalent production method to prepare ultra-thin grain-oriented silicon steel is to use commercial grain-oriented silicon steel sheets as starting material, then cold-rolling the steel sheets to the desired thickness followed by annealing processes [4,10,11].

In recent years, many studies have reported the formation of deformation twinning in silicon steel. Shi et al. [12] and Xie et al. [13] discovered that deformation twinning occurred in Fe-6.5% Si alloy in a medium temperature tensile and compress test, and that this twinning promoted the plastic deformation of the alloy. Dunn et al. [14] demonstrated that both slip and twinning were activated during cold rolling in Fe-3.25 wt.% Si alloy, and {001}<110> oriented twins could be formed in Goss single crystal at an early deformation stage. Rusakov et al. [15] studied the features of twinning in cold-deformed Goss single crystal in an Fe-3% Si-0.5% Cu alloy and found that twins with near {001}<110> orientation were formed at 5% reduction and the twinning orientation did not change during the subsequent deformation. Dorner et al. [16] reported that the area fraction of {001}<110> oriented

twins increased with an increase of deformation reduction and reached the maximum at 61% deformation reduction when studying the evolutions of crystallographic orientations of cold-rolled Goss single crystals in Fe-3% Si alloy. Even with extensive studies, there are still disputes on how the twins in BCC structured grain-oriented silicon steels are formed and evolved during cold rolling.

It is known that deformation twinning has a pronounced grain orientation dependence in FCC [17–19] and HCP [20,21] metals or alloys, and similar effects of initial grain orientation on deformation twinning are also reported in BCC structure [22,23]. Fu et al. [22] found that twins tend to occur in grains with a tensile orientation near the <001> corner and a compressive orientation near the <101>-<111> line, and this twinning activation is closely related to their corresponding Schmid factor, respectively. In the production of ultra-thin grain-oriented silicon steel, it is preferred that the starting material has a strong Goss texture, so that grains with exact Goss in a certain extent in the resultant UTGO steel will occupy the great majority of area. The deviation of grain orientation from the exact Goss will affect the subsequent orientation transition routes [23]. However, the effect of deviation degree on the twinning behavior remains unclear. In this study, the twinning behaviors during cold rolling ultra-thin grain-oriented silicon steel have been systematically analyzed, and special attention has been focused on the influence of initial Goss orientation deviation on the twinning behavior. The results will help to comprehensively understand the cold rolling process and to provide a theoretical basis for preparing ultra-thin grain-oriented silicon steel.

## 2. Experimental Procedure

A 0.35 mm-thick commercial glassless grain-oriented silicon steel plate without a magnesium silicate layer was used as raw material to prepare the ultra-thin silicon steel. Its magnetic properties were $B_8$ = 1.89 T, $P_{1.7/50}$ = 1.12 W/kg, the average grain size was about 30 mm and the chemical compositions (mass fraction, %) were C: 0.0058, Si: 3, Mn: 0.0088, S: 0.0003, Al: 0.005, N: 0.001, Sn: 0.1. After being pickled with hydrochloric acid to distinguish the original grain boundaries, the sample plates of 300 mm (along rolling direction, RD) × 40 mm (along transverse direction, TD) were cut into two parts, as shown in Figure 1. The parts marked by the red block diagram are 10 mm (RD) × 5 mm (TD) and were used to confirm the initial orientation of the marked grains. The analysis was done using a Zeiss GeminiSEM500 field emission scanning electron microscope (SEM) equipped with an EBSD system and the accelerating voltage was 20 KV when measured. EBSD data was post-processed by the HKL CHANNEL5 software. The part marked by the green block is 300 mm (RD, rolling direction) ×30 mm (TD, transverse direction) and was cold rolled to 0.21–0.075 mm thick to investigate the twinning behavior. The microstructures and microtextures at different reductions were characterized and evaluated with a Confocal laser scanning microscope (CLSM) and EBSD technique, respectively. For the EBSD measurements, the samples were machined and electropolished with a 5% perchloric acid/alcohol solution to remove the surface strain layer. In the EBSD data, the tolerance angles for the orientations are set as 15°.

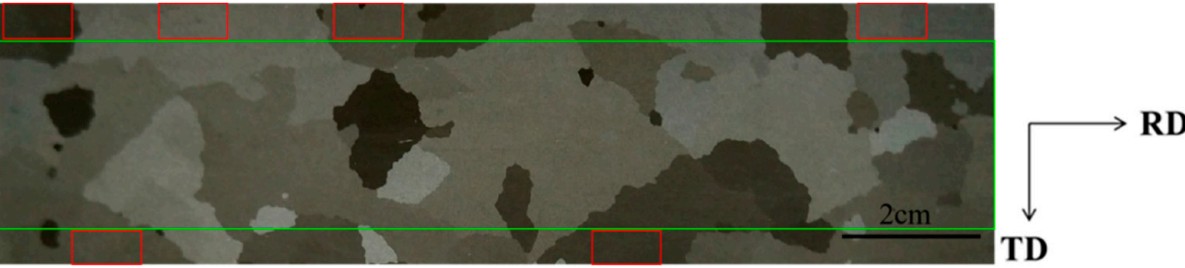

**Figure 1.** Schematic diagram of sample cutting.

## 3. Results and Discussion

### 3.1. Deformation Twinning Microstructure

The microstructures of the 0.075 mm rolled sheet were characterized by CLSM and are shown in Figure 2, and the textures measured by EBSD are presented in Figure 3. Parallel bands with serrated edge and different angles toward RD direction were found. The measurement shows the parallel bands are {001}<110> orientation and they are in <111> /60° relationship with the {111}<112> oriented matrix.

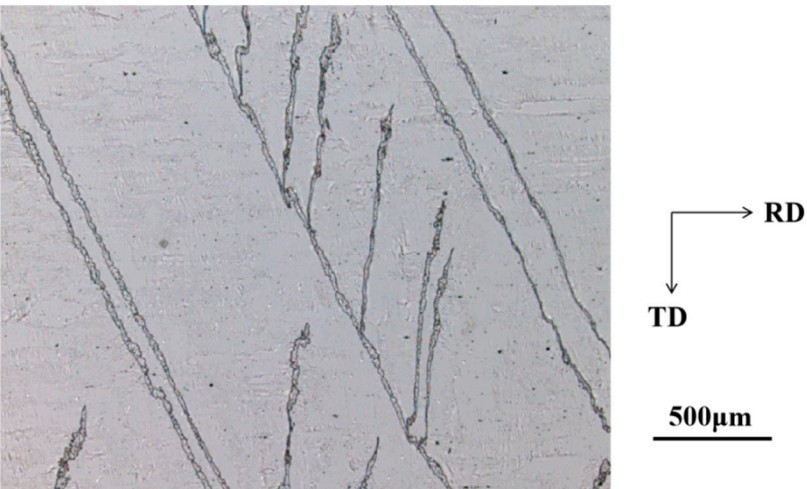

**Figure 2.** SEM micrographs in deformation twinning area of 0.075 mm sheet in rolling plane.

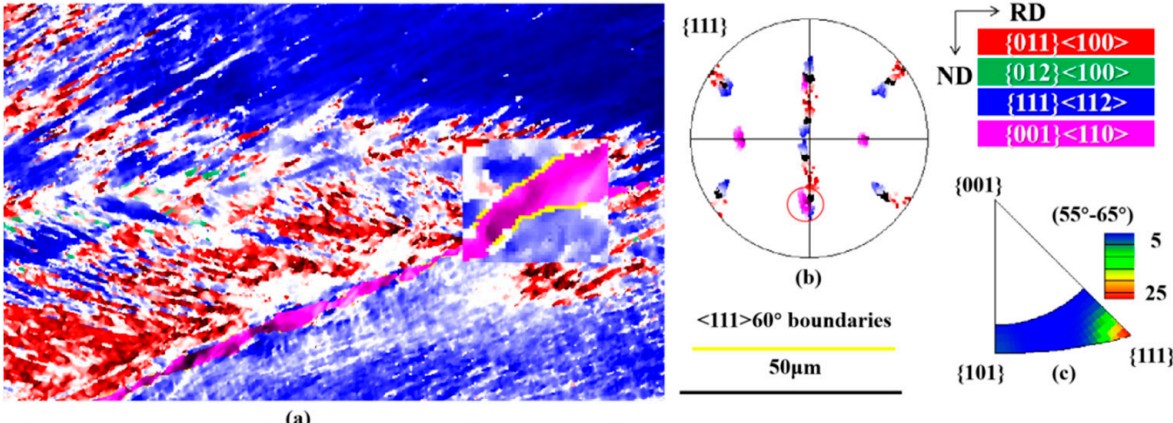

**Figure 3.** EBSD measurement of 0.075 mm sheet in transverse plane with step size = 0.2 μm: (**a**) EBSD orientation image mapping; (**b**) {111} pole figure; (**c**) rotation axes distribution for misorientation angle of 55–65° in (**a**).

### 3.2. Twinning Behavior in Goss-Oriented Grains during Cold Rolling

In this paper, the twinning Schmid factor and shear mechanical work of cold rolled Goss oriented grains at different strains are calculated and the effect of grain orientation on the selection of the twinning system is discussed [24–26]. The evolution of the twinning behavior during cold rolling in ultra-thin grain-oriented silicon steel is investigated by theoretical calculation and is verified via experiments.

The microstructures of the grains with a 3° angle deviated from exact Goss orientation at different reductions are depicted in Figure 4. The microstructure and microtextures at 78% reduction measured with EBSD are shown in Figure 5. With the increase of deformation reduction, the area fraction of twins increases. When the deformation reduction reaches 78%, the edge serration of the twins becomes more prominent. Twins with three different

deviation angles from exact {001} <110> orientation are observed in Figure 5a, and it is suggested that they were formed at different deformation stages during cold-rolling.

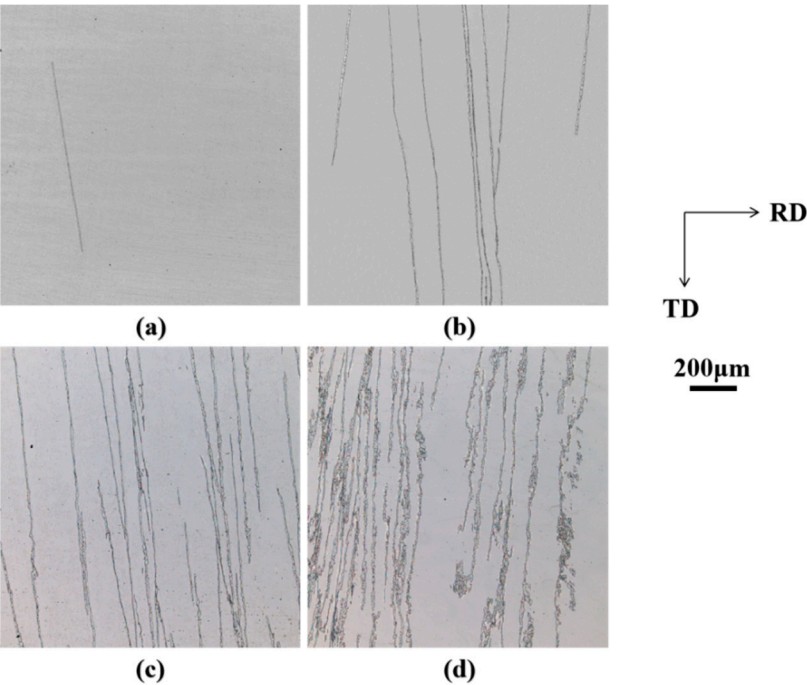

**Figure 4.** The morphology of deformation twinning within Goss-oriented grains at different reductions in rolling plane. (**a**) 32%; (**b**) 41%; (**c**) 53%; (**d**) 78% reduction.

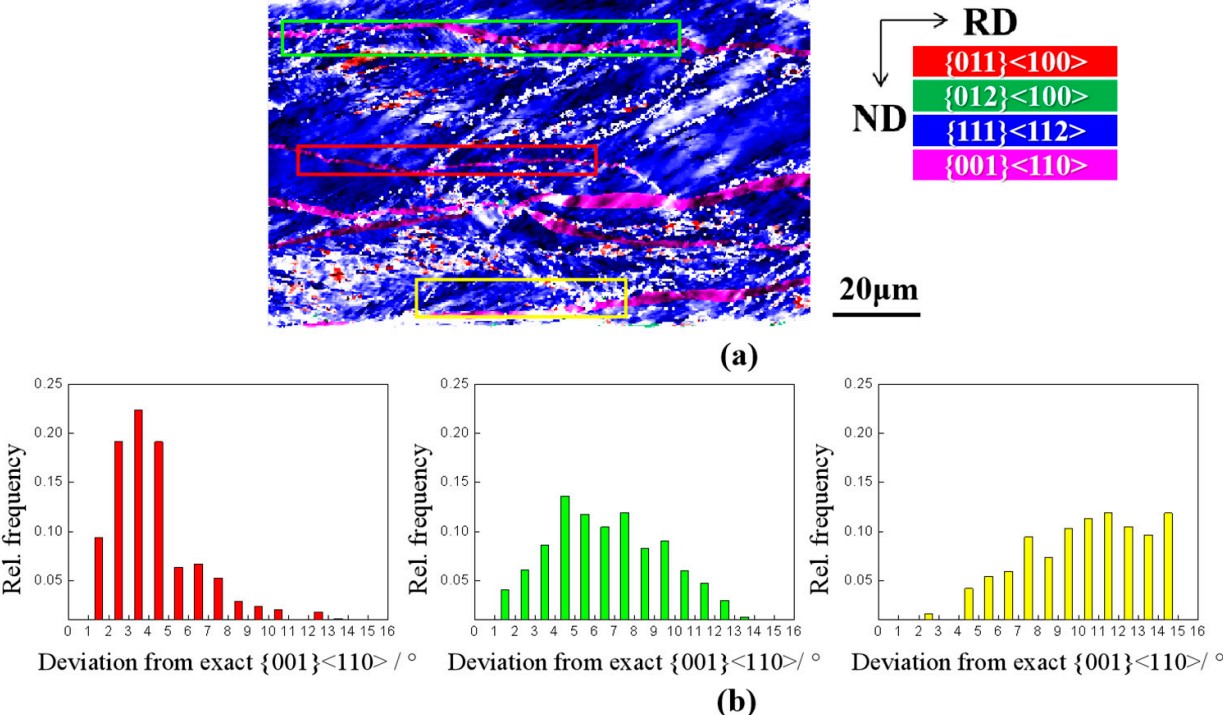

**Figure 5.** EBSD measurement in deformation twinning area of exact Goss orientated grains with a reduction of 78% in transverse plane with step size = 0.3 μm: (**a**) orientation image mapping; (**b**) the distribution of the deviation degree between twinning orientation and exact Goss orientation in the area marked by different color block diagram in (**a**).

According to Humbert's observation [24], the deformation twinning in BCC alloy can be obtained when every two adjoining (112) planes are displaced toward [111] direction by $\sqrt{3}a/6$. By these successive displacements, the BCC lattice is sheared by $\sqrt{2}/4$.

In the reference frame $Y_T$ based on the corresponding twinning system, this plane strain is expressed by matrix $E_T$.

$$
Ey = \begin{bmatrix} 0 & 0 & \frac{\sqrt{2}}{4} \\ 0 & 0 & 0 \\ \frac{\sqrt{2}}{4} & 0 & 0 \end{bmatrix}
\tag{1}
$$

In the reference frame based on deformed matrix orientation, the twinning strain matrix, E, is expressed as

$$
E = N \cdot M \cdot E_T \cdot M^{-1} \cdot N^{-1}
\tag{2}
$$

Herein, M and N are corresponding transformation matrices of the twinning strain matrix from the reference frame based on corresponding twinning system to sample coordinate system and crystal coordinate system based on deformed matrix orientation successively, and the corresponding mechanical work of the twinning shear is calculated as

$$
W = \frac{1}{2}(\varepsilon_{11}\cdot E_{11} - \varepsilon_{33}\cdot E_{33}) = \frac{1}{2}\varepsilon_{11}(E_{11} - E_{33})
\tag{3}
$$

Herein, the rolling stress is simplified as equivalent force in the direction of ND and RD, that is, $\varepsilon 11 = \varepsilon 33$.

The twinning forming stages can be deduced based on the orientation relationship between twinning and deformed matrix. On the other hand, the selection of the twinning system can be analyzed based on the orientation of the deformed matrix when twinning occurs. The VPSC (visco-plastic self-consistent) model has been applied to predict the orientation rotation path of Goss oriented grain during cold rolling and the result is shown in Figure 6.

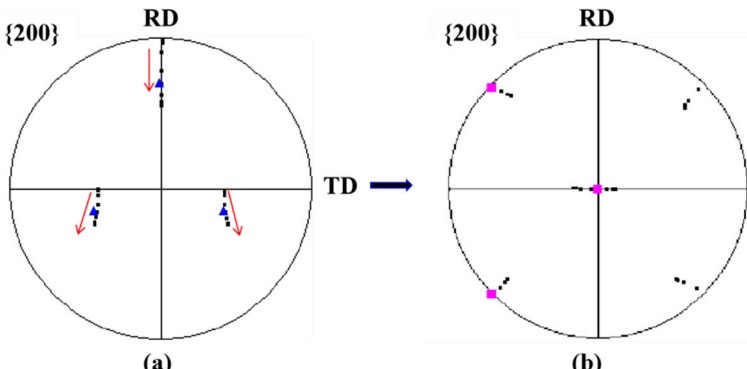

▲    Matrix orientation when strain is −0.6

■    Twin orientation when strain is −0.6

**Figure 6.** The orientation rotation path of Goss orientation: (**a**) corresponding twinning orientation; (**b**) during cold rolling, strain value = −0.6.

The calculation results suggest that during cold rolling, Goss orientation rotates to {111}<112> orientation along the route indicated by red arrows in Figure 6a, which is consistent with the literature [27]. In the calculated deformed matrix, the twinning orientations can be predicted based on the maximum Schmid factor and shear mechanical work of the twinning systems, as shown in Figure 6b. The orientation of the twinning forming during the cold rolling of Goss oriented grains is always α-fiber, close to {001}<110> orientation but with different degree of deviation. This is in good agreement with the

experimental results, suggesting that twinning can occur in Goss oriented grains across a cold rolling process. The twinning orientation of exactly {001}<110> is achieved when the strain is about –0.6 and the deformed matrix orientation is (18° 48° 153°). Since most twins have orientations of exact or close to {001}<110>, we predict that twins mainly formed at this strain.

In a matrix of (18°, 48°, 153°) orientation at strain equal to –0.6, the Schmid factor and shear mechanical work of the twinning systems and their corresponding twinning orientations were estimated and are listed in Table 1. It is obvious that the Schmid factor and shear mechanical work of the twelve twinning systems increase at the same time; in other words, the maximum Schmid factor and shear mechanical work will correspond to the same twinning system, which leads to the twinning orientation of (136° 90° 90°). Similarly, the same phenomenon applies to the deformed orientations at different stains, and the calculated twinning orientations are always close to or near {001}<100>. The results confirm the assumption that the activated twinning system during cold rolling of Goss orientation grain always has the maximum value of Schmid factor and shear mechanical work. The greater the Schmid factor and shear mechanical work, the easier a twinning system would occur.

**Table 1.** The Schmid factor, shear mechanical work of twins in the deformed matrix with orientation of (18° 48° 153°).

| Twinning System | Schmidt Factor | Mechanical Work | Twinning Orientation |
|---|---|---|---|
| $(112)[11\bar{1}]$ | 0.48 | $0.167\varepsilon_{33}$ | (46° 90° 0°) |
| $(121)[1\bar{1}1]$ | 0.26 | $0.093\varepsilon_{33}$ | (90° 141° 45°) |
| $(211)[1\bar{1}\bar{1}]$ | 0.36 | $0.130\varepsilon_{33}$ | (106° 116° 8°) |
| $(\bar{1}21)[\bar{1}1\bar{1}]$ | 0.26 | $0.093\varepsilon_{33}$ | (37° 116° 30°) |
| $(1\bar{1}2)[\bar{1}111]$ | 0.14 | $0.046\varepsilon_{33}$ | (122° 27° 105°) |
| $(11\bar{2})[111]$ | 0.52 | $0.176\varepsilon_{33}$ | (105° 64° 83°) |
| $(\bar{1}21)[\bar{1}\bar{1}1]$ | 0.46 | $0.167\varepsilon_{33}$ | (42° 180° 86°) |
| $(1\bar{2}1)[\bar{1}\bar{1}\bar{1}]$ | 0.14 | $0.046\varepsilon_{33}$ | (162° 84° 154°) |
| $(12\bar{1})[1\bar{1}\bar{1}]$ | 0.50 | $0.176\varepsilon_{33}$ | (161° 97° 116°) |
| $(\bar{2}11)[\bar{1}\bar{1}\bar{1}]$ | 0.38 | $0.130\varepsilon_{33}$ | (120° 153° 14°) |
| $(2\bar{1}1)[11\bar{1}]$ | 0.94 | $0.333\varepsilon_{33}$ | (136° 90° 90°) |
| $(21\bar{1})[1\bar{1}1]$ | 0.52 | $0.185\varepsilon_{33}$ | (141° 117° 60°) |

*3.3. Effect of Goss Orientation Accuracy of Initial Grain on Twinning Behavior*

The majority of grains in the raw material have orientations close to Goss with 3°–10° deviation. The microstructures of 3° and 10° deviated Goss grains at 78% reduction were characterized by metalloscopy in the ND plane and by EBSD in the TD plane. The typical images are shown in Figures 7 and 8, respectively. Compared to 10° deviated Goss oriented grains, there is much higher density of twins in 3° deviated Goss oriented grains and the twinning orientations are more closely aligned to the exact {001}<110> orientation.

The crystal rotation routes of 10° deviated Goss orientation (0° 35° 0°) during cold rolling were simulated using VPSC model. Meanwhile, the Schmid factor and shear mechanical work as well as twinning orientation of the twinning systems were calculated in matrices with various orientations at different strain values. The results are shown in Figure 9 and Table 2. During cold rolling, the crystal rotation routes of grains with deviated Goss orientation are similar to that of exact Goss grains, showing deviated {111}<112> orientation at 78% rolling reduction. Moreover, under the same strain, the twining Schmid factor and shear mechanical work in deviated grains are both lower than that of exact Goss oriented grains. In a deviated Goss grain, the twins generally have a deviation angle with the exact {001}<110> orientation. The relationship of this deviation angle with {001}<110> orientation and strain is analyzed and the simulated results are presented in Figure 10. At the same strain, the twins in the initial deviated Goss grains have twinning orientation further away from the exact {001} <110> orientation. Since twinning systems

are critical for achieving the final microstructures in ultra-thin grain-oriented silicon steel, the deviation degree of initial Goss oriented grain, therefore, would influence the evolution of the microstructure and microtextures of recrystallized steel. This influence is subject to future study.

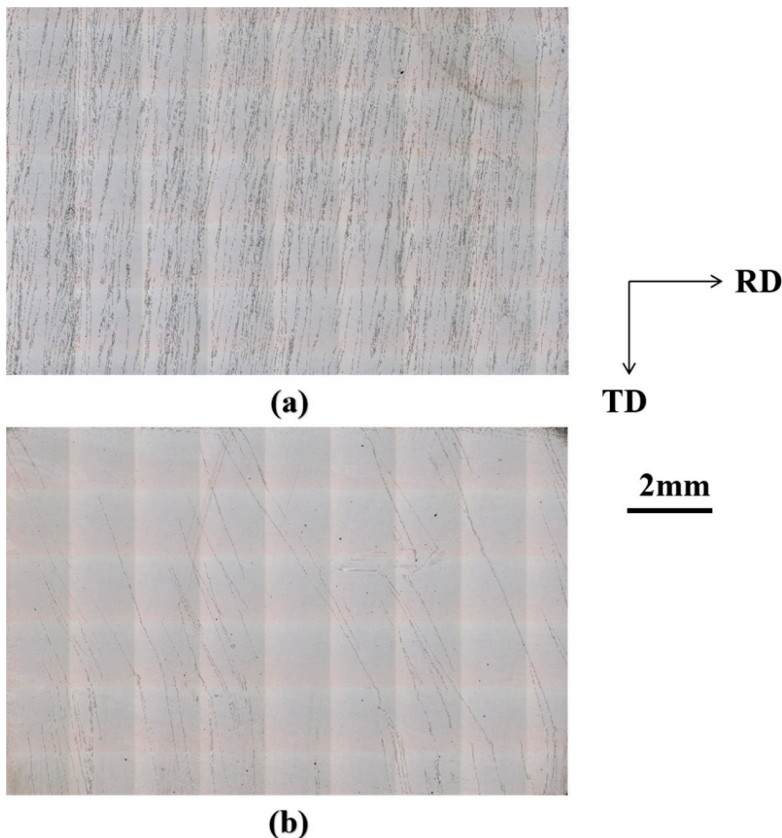

**Figure 7.** The effect of the deviation degree of the initial Goss orientation on twinning distribution in rolling plane: (**a**) 3° deviated Goss oriented grain; (**b**) 10° deviated Goss oriented grain.

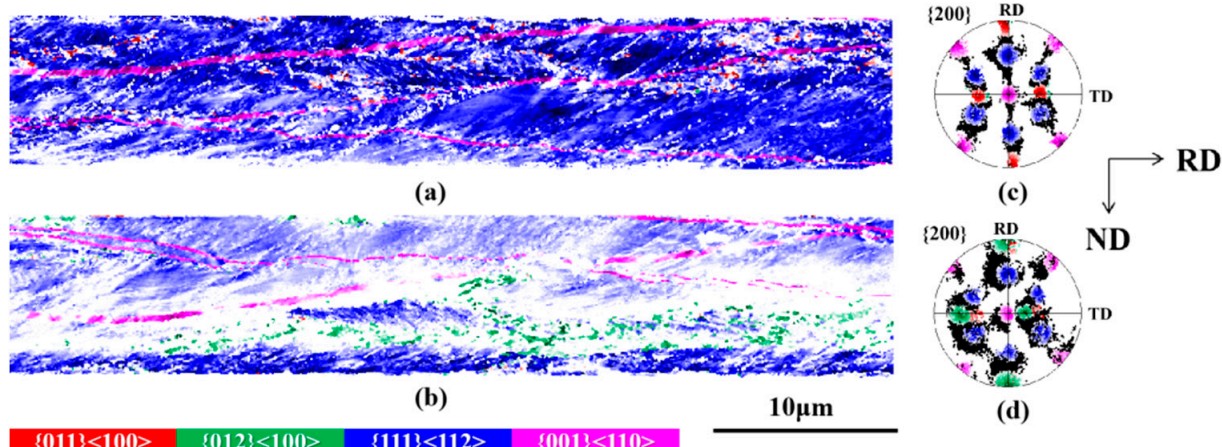

**Figure 8.** The effect of deviation angle of Goss oriented grain on twinning orientation in the lateral plane: (**a**) 3° deviated Goss oriented grain; (**b**) 10° deviated Goss oriented grain; EBSD measurement step size = 0.5 μm.

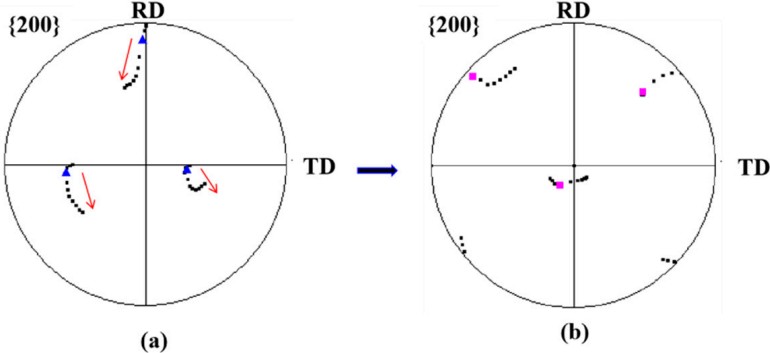

▲   Matrix orientation when strain is −0.6

■   Twin orientation when strain is −0.6

**Figure 9.** The orientation rotation path of (0° 35° 0°) orientation; (**a**) and corresponding twinning orientation (**b**) during cold rolling, strain = −0.6.

**Table 2.** Schmid factor and shear mechanical work of (0° 45° 0°) orientation and (0° 35° 0°) orientation at different strain in cold rolling (MSF and MMW represent maximum Schmid factor and shear mechanical work, respectively).

| Strain | MSF of (0° 45° 0°) | MMW of (0° 45° 0°) | MSF of (0° 35° 0°) | MMW of (0° 35° 0°) |
|---|---|---|---|---|
| 0 | 0.951 | $0.336\varepsilon_{33}$ | 0.935 | $0.329\varepsilon_{33}$ |
| −0.2 | 0.962 | $0.340\varepsilon_{33}$ | 0.938 | $0.331\varepsilon_{33}$ |
| −0.4 | 0.989 | $0.350\varepsilon_{33}$ | 0.948 | $0.336\varepsilon_{33}$ |
| −0.6 | 0.988 | $0.350\varepsilon_{33}$ | 0.965 | $0.341\varepsilon_{33}$ |
| −0.8 | 0.929 | $0.329\varepsilon_{33}$ | 0.952 | $0.337\varepsilon_{33}$ |
| −1.0 | 0.845 | $0.298\varepsilon_{33}$ | 0.887 | $0.313\varepsilon_{33}$ |
| −1.2 | 0.756 | $0.269\varepsilon_{33}$ | 0.823 | $0.288\varepsilon_{33}$ |
| −1.4 | 0.714 | $0.254\varepsilon_{33}$ | 0.776 | $0.263\varepsilon_{33}$ |
| −1.6 | 0.710 | $0.251\varepsilon_{33}$ | 0.739 | $0.242\varepsilon_{33}$ |

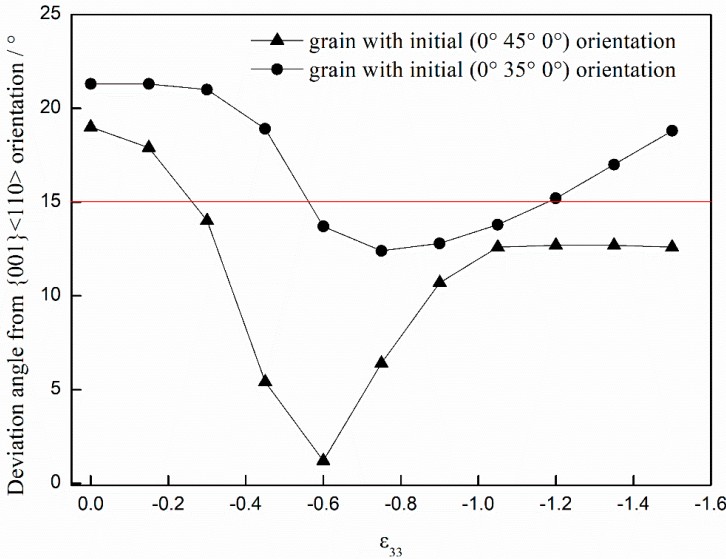

**Figure 10.** The effect of initial grain orientation on the deviation angle of twinning orientation from exact {001}<110> orientation during cold rolling.

## 4. Conclusions

The actual activated {112}<111> twinning systems are shown to have the maximum Schmid factor and shear mechanical work. The greater Schmid factor and shear mechanical work a twinning system has achieved, the easier it is to be activated.

Twinning can occur at different stages of rolling deformation in Goss oriented grain during cold rolling. The twinning area fraction increases with the increase of rolling reduction, and the twinning orientations are mainly located at α-fiber and are mostly around {001}<110> component. Based on VPSC simulation results, exact {001}<110> oriented twinning is achieved when the rolling reduction is 40–50%.

In the preparation of ultra-thin grain-oriented silicon steel, when an initial material has more grains with exact Goss orientation, more twins are observed in the deformed grains after cold-rolling and the twinning orientation is closer to exact {001}<110>.

**Author Contributions:** Conceptualization, L.M.; Data curation, B.Z.; Methodology, L.M.; Project administration, L.M.; Resources, G.M. and K.L.; Supervision, N.Z. and G.L.; Writing—original draft, B.Z.; Writing—review & editing, N.Z. and S.Z. All authors have read and agreed to the published version of the manuscript.

**Funding:** National Key Research and Development Program of China: 2016YFB0300301, 2017YFB0903901; Shanxi Provincial Science and Technology Major Special Project: 20191102004.

**Acknowledgments:** The authors gratefully acknowledge the funding of this work by the National Key Research and Development Program of China (Grant Nos. 2016YFB0300301 and 2017YFB0903901) and the Shanxi Provincial Science and Technology Major Special Project (20191102004).

**Conflicts of Interest:** The authors declare no conflict of interest.

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
