# Peer review of "Twinning Behavior in Cold-Rolling Ultra-Thin Grain-Oriented Silicon Steel"

_crystals, doi:10.3390/cryst11020187_

Round 1

Reviewer 1 Report

Twinning behavior in UTGO steel has been studied in this study by conducting EBSD analysis and deformation orientation simulation. The interpretation from the experimental results is well written logically and I think it is reasonable to publish the paper in this journal. I have a few minor questions and suggestions.

In EBSD analysis, orientation detection is considered to be difficult as deformation reduction increases. In Figure 8(b) of orientation image map, the white region is may be undetected area. The difficulty of orientation measurement at high degree of deformation is likely to affect accurate analysis of twinning behavior. Perhaps consideration of this is needed. In experimental section, detailed EBSD measurement condition should be added.

Reviewer 2 Report

The manuscript titled Twinning Behavior In Cold-Rolling Ultra-Thin Grain-Oriented Silicon Steel of authors Bo Zhang et al. is very interesting. The topic is highly relevant and very important for the scientific field. However, some shortcomings have been identified, as follows:

Title: Cold-Rolling or maybe Cold-rolled? 

  1. Experimental procedure:

Missing:

- chemical composition of the steel;

- preparation procedure for EBSD analysis;

- more detailed parameters of FE-SEM/EBSD should be provided (accelerating voltage, step size, etc.)

Add, please.
